# Adaptive Remote Sensing Paradigm for Real-Time Alerting of Convulsive Epileptic Seizures

**DOI:** 10.3390/s23020968

**Published:** 2023-01-14

**Authors:** Stiliyan Kalitzin

**Affiliations:** Stichting Epilepsie Instellingen Nederland (SEIN), 2103 SW Heemstede, The Netherlands; skalitzin@sein.nl; Tel.: +31-235588248

**Keywords:** epilepsy, convulsive seizures, optical flow, video processing, adaptive algorithms, unsupervised learning

## Abstract

Epilepsy is a debilitating neurological condition characterized by intermittent paroxysmal states called fits or seizures. Especially, the major motor seizures of a convulsive nature, such as tonic–clonic seizures, can cause aggravating consequences. Timely alerting for these convulsive epileptic states can therefore prevent numerous complications, during, or following the fit. Based on our previous research, a non-contact method using automated video camera observation and optical flow analysis underwent field trials in clinical settings. Here, we propose a novel adaptive learning paradigm for optimization of the seizure detection algorithm in each individual application. The main objective of the study was to minimize the false detection rate while avoiding undetected seizures. The system continuously updated detection parameters retrospectively using the data from the generated alerts. The system can be used under supervision or, alternatively, through autonomous validation of the alerts. In the latter case, the system achieved self-adaptive, unsupervised learning functionality. The method showed improvement of the detector performance due to the learning algorithm. This functionality provided a personalized seizure alerting device that adapted to the specific patient and environment. The system can operate in a fully automated mode, still allowing human observer to monitor and override the decision process while the algorithm provides suggestions as an expert system.

## 1. Introduction

Epilepsy is a disorder of the central nervous system where intermittent, and in general, unpredictable transitions to paroxysmal states, called seizures, can occur. These states elicit synchronous activity of the brain; they can recruit different areas of brain tissue and have a variety of clinical manifestations. In some cases, the epileptic condition is benign but in more severe forms of the disorder, patients have a significant risk of injuries, and even death, as a result of the seizures [1,2]. Especially vulnerable are the sufferers that experience convulsive fits or seizures. During those events, timely assistance can be critical for the health and wellbeing of the patient. Several products on the market provide alerts for convulsive seizures [3,4]. Some use wearable sensors typically attached to the patient’s arm, others register vibrations of the bed and alert for nocturnal fits. In all these products, direct or indirect contact between the patient and the sensors is required for the proper operating of the system. Such systems are therefore sensitive to accidental or deliberate misplacement of the primary sensor. In the case of wearable sensors, regular charging or battery change is also necessary. All these requirements make the use of contact sensors in some cases difficult, and may lead to additional workload from the caregivers. In many care facilities, continuous video monitoring is used for the safety of the patients. This involves constant time-consuming alertness of the personnel and may compromise the privacy of the patients. 

To address the above issues in our care facility, we have developed a system for remote sensing and alerting for convulsive epileptic seizures. The system uses a continuous stream of images from a video camera providing a relatively constant rate of frame acquisition. During dark periods, the camera automatically switches to active infrared enhanced acquisition. Proprietary algorithms process, in real time, the image sequence and upon certain conditions may issue an alert. A standard nursing alert system receives the signal of an epileptic event and dispatches it to the caregiver’s observation post, and, if required, to mobile devices as well. We explain in detail the overall concept in the next section. The processing algorithms were tested earlier on pre-recorded video sequences screened by clinical experts [5]. Results from night observations in a care facility [6] and in children at home [7] have proven that video-based seizure detection can provide sufficient effectiveness and security. As it is a non-contact, non-obstructive approach, it can also operate in combination with other detection modalities [8], depending on the particular situation. 

In short, our method uses optical flow type of reconstructive analysis that infers the rates of changes of a limited number of group transformation parameters, such as translations, rotations, dilatations, and shear transformations, from the video sequence. Subsequently, the obtained signals undergo time-frequency filtering using wavelet decomposition. Finally, we analyze the filtered signal that represents the likelihood of a convulsive seizure and upon pre-defined conditions produce an alert signal that deploys directly via a standard “dry contact” interface to an existing nurse alerting system. 

The essential innovation proposed in this contribution is the addition of an adaptive or machine learning (ML) functionality to the system. We achieved this on two levels, by tuning the decision parameters for raising an alert, and on a deeper level, by dynamic selection of the wavelet filter parameters. We have realized an on-going system adaptation procedure that operates in parallel with the uninterrupted functioning of the alerting process. Unlike other ML paradigms, we do not require a training set of data to be available prior to the initial operational state. Our system acquires its own set of events and uses them to improve its performance. We introduce two possible scenarios of application. The first option relies on a supervised validation of the detected events where a qualified observer identifies the true detections from the false alerts. This involves time-consuming visual inspection on the recorded video fragments associated with the alarms. Alternatively, an automated off-line classification is proposed. In this case, the system works completely autonomously. When required, the operator’s inspection only provides an assessment of the system performance. We achieved the automated classification by estimating the average distance to similar events surrounding each detected event. Assuming that epileptic movements are compulsive and therefore more stereotyped than the “normal” movements of the body, the average distance provides a simple but effective clustering criterion that separates true from false detections. The distance metric uses the total optical flow information collected during the system operation. To summarize, the novelties proposed here are: (1) introduction and implementation of an algorithm for continuous adaptation of the seizure detection parameters; and (2) introduction and implementation of an autonomous procedure for retrospective validation of the detected events. In addition, this work implements a real-time version of a previously published original algorithm for optical flow group velocities reconstruction.

The organization of the rest of the paper is as follows. In Section 2, Section 2.1, Section 2.2 and Section 2.3 we present the clinical case considered as well as the overall layout of the seizure detection system and brief technical details related to the hardware as implemented. Section 2.4, Section 2.5 and Section 2.6 describe the “rigid” version of the algorithm comprising optical flow reconstruction of group parameters, wavelet filtering and the alert decision procedure. Section 2.7 introduces the novel approach of frequency interval selection as part of our adaptive algorithm. Section 2.8 presents our decision parameter optimization approach. Section 2.9 contains the formulation of our novel procedure for unsupervised retrospective validation of detected events based on clustering technique. Section 3 presents our results from the trial. First in Section 3.1 we show the detection statistics of the default “rigid” algorithm. Section 3.2 contains the results obtained after applying supervised parameter optimization. In Section 3.3, we introduce results from the autonomous, unsupervised adaptive algorithm. In addition to the detection statistics, we provide assessment of the automated event classification from two separate sequences of events during the trial. Section 4 offers a discussion of our overall strategy, open problems, limitation and possible future directions of our quest for a reliable seizure alerting device.. Finally, we summarize our conclusions in Section 5. 

## 2. Materials and Methods

### 2.1. Patient Data

In this technical communication, we present results from a case study of single patient data as a proof-of-concept. The subject, a 46-year-old male, suffered from pharmacologically intractable epilepsy with a spectrum of motor disturbances. In addition to the tonic–clonic seizures of an average frequency of 1.5/day, the patient elicited successions of short convulsive movements at irregular periods during sleep. He also experienced impaired arm movement due to the frequent tremors. The above conditions make the selective detection of only the major motor TC seizures a significant challenge. Most detectors cannot distinguish easily between the various motor paroxysms. 

### 2.2. Processing Flow

The overall processing flow and system layout shown on Figure 1 represents both the supervised (panel a) and unsupervised (b) operational modes. 

Deployable application written on Matlab^®^, version 2022b (The Mathworks Inc., Natick, MA, USA) incorporates all software modules and routines. The application runs on a HP Pavilion Desktop 595 personal computer equipped with 3GHz Intel I7 quad-core processor, 16Gb internal memory. The operational system is Windows 10 Home, version 22H2. Parallel processing for matrix calculus uses a NVIDEA GeForce GTX 1050 graphic card. For the connectivity to the nurse alerting system (De Heer Medicom^®^) we use DLP-IOR4 (DLP Design Inc., McKinney, TX, USA) USB based latching relay module operating in Normal-Open protocol. 

### 2.3. Video Acquisition

In the “field trial” setting, we used a USB camera (720P USB2.0 OmniVision OV9712 Color CMOS Sensor USB Camera, AILIPU TECHNOLOGY CO., LTD., Shenzhen, China) that provides fixed frame rate and day-night infrared LED capabilities. The frame rate was constant at 24 frames per minute; resolution was 640 by 480 pixels, subsequently down sampled to 320 by 240. To this end, we applied a 2 × 2-pixel block averaging to reduce the image resolution. The post-acquisition software processes image sequences of given length in sequential cycles (in this work we used 36 frames per cycle corresponding to 1.5 s). For each processing cycle, an optical flow reconstruction algorithm explained in brief below applies to each consecutive pair of frames. The acquisition cycles of image sequences follow each other continuously but their processing is mutually independent. 

### 2.4. Optical Flow Group Velocities Reconstruction 

The theory and proprietary algorithm used to derive motion rates from image sequences are introduced in [9,10]. Here we recall the main idea. 

Optical flow is an image-processing algorithm that aims at reconstructing the velocities of moving objects from the time-changes in the sequences of video images taken from those objects. If we are interested in only certain overall global movement rates, in our case, those of the two translations, rotation, dilatation and the two shear transformations, we can apply a direct reconstruction algorithm that saves a lot of computational power and complexity. Compared to other optical flow techniques, we avoid the reconstruction of local velocities vector field in all image locations (pixels). As a result, applying the algorithm on the image sequence produces six time series representing the rates of changes (group velocities) of the six two-dimensional linear inhomogeneous transformations.
(1)$$L^{c}\left(x,y,t\right);c=\left\{R,G,B\right\}=>V_{g}\left(t\right);g=\left\{TrX,\;TrY,\;Rot,\;Dil,\;ShX,ShY\right\}$$

Our algorithm uses all the available spectral components of the video sequence (for our case just red, green, and blue components), in contrast to other available algorithms that process only the intensity values in the images. 

### 2.5. Wavelet Decomposition and Filtering. Seizure Biomarker

We use a set of Gabor wavelets with exponentially increasing wavelengths $f_{k},\;k=1\ldots 200$
$$W_{g}\left(t,f_{k}\right)=\left|\int_{t^{\prime }}\;dt^{\prime }G\left(t-t^{\prime },f_{k}\right)V_{g}\left(t^{\prime }\right)\right|\;$$
$$W\left(t,f_{k}\right)\equiv \langle W_{g}\left(t,f_{k}\right)\rangle_{g}$$
(2)$$W^{q}\left(f_{k}\right)\equiv \langle W\left(t,f_{k}\right)\rangle_{t\in q}$$

For the exact definitions and normalizations, we refer to earlier publications [3] and here we note that the wavelet spectrum in (2) is a time average along each image sequence denoted with q. 

Next, we define the “epileptic content” as the fraction of the wavelet energy contained in the frequency range defined here as $\left[f_{a},f_{b}\right]$.
(3)$$E\left(q,f_{a},f_{b}\right)\equiv \frac{\sum_{f\in \left[f_{a},f_{b}\right]}W^{q}\left(f\right)}{\sum_{f}W^{q}\left(f\right)}$$

In the “rigid” application, as well as an initial setting for the adaptive scheme, we use f∈[2…7] Hz. To compensate for different frequency ranges that may be used, we calculated also the same quantity in (3) but for a signal with “flat” spectrum representing random noisy input. Then we rescale the epileptic marker as
(4)$$\overset{ˇ}{E}\left(q,f_{a},f_{b}\right)=\frac{E\left(q,f_{a},f_{b}\right)-E_{0}\left(f_{a},f_{b}\right)}{1-E_{0}\left(f_{a},f_{b}\right)}$$

Here $E_{0}$ is the relative wavelet spectral power of a white noise. 

Note that in (3) the quantity q is a discrete index representing the frame sequence number and corresponds, as stated earlier, to a time window of approximately 1.5 s. 

### 2.6. Event Detection 

We use three parameters [N,n,T] to detect an event (seizure alert) in real time. At each time instance, we take the seizure marker (4) in the N preceding windows. If from those, N, or at least n have values $\overset{ˇ}{E}\;$> T, an event is generated and eventually (if within the time selected for alerts) sent as an alert. The default values are [7 6 0.4]. This corresponds to a criterion that detects if from the past 10.5 s at least 9 s contain an epileptic “charge” (4) higher than 0.4. These are the values for the rigid mode as well as for the initial setting in the adaptive mode. 

Figure 2 is an illustration of the event detection algorithm. 

### 2.7. Adaptive Frequency Range Selection

If we have a validated set {S} of seizure events, we can estimate the relevant frequency range $\left[f_{1},f_{2}\right]$ used in Equations (3) and (4) as follows. 

First, we average the frequency mean-subtracted spectrum over a number of sequences q around the seizure events
(5)$$K\left(f_{k}\right)=\langle {\tilde{W}}^{q}\left(f_{k}\right)\rangle_{s,q}\frac{\left|\langle {\tilde{W}}^{q}\left(f_{k}\right)\rangle_{s,q}\right|}{\left|\langle {\tilde{W}}^{q}\left(f_{k}\right)\rangle_{s,q}\right|+std_{s}\left(\langle {\tilde{W}}^{q}\left(f_{k}\right)\rangle_{q}\right)}$$

Here we define ${\tilde{W}}^{q}\left(\nu_{k}\right)=W^{q}\left(\nu_{k}\right)-W^{q}{\left(\nu_{k}\right)}_{k}$

In Equation (5) subscript s is the seizure event label. We also introduce a “penalty” for high variability around the average values. Conveniently, we take 10 windows before and 15 windows after each event for the time averaging over q. 

Next, we determine the optimal frequency interval for seizure event detection as
(6)[fa,fb]={f,K(j)maxf(K(f))>0.1}

### 2.8. Selection of Optimal Parameters for Event Detection

Once we have the biomarker for epileptic movements from Equation (4) which is calculated from either the default or the “trained” frequency range (6), we can optimize the decision triple of parameters {N,n,Tt} by introducing the following cost-function
C(T,N)=T(2n(T,N)−N)
(7)$$n\left(T,N\right)\equiv min_{S|T,N}\left(n\right)$$

In the second equation, the value of the parameter n, the filling number, is derived for each pair of (T,N) values as the minimal number of threshold-exceeding epochs n over all detected and validated seizure events. This way we can guarantee that, for any choice of the pair (T,N), all the previously detected true seizure events will be preserved. Therefore, we can postulate the optimized choice of parameters as:{T,N}=argmax(C(T,N));
(8)n=n(T,N) 

### 2.9. Unsupervised Validation of Event Detection

Here we recall the original optical flow reconstruction representation (1) for each video frame sequence *q*. 

Our main assumption is that epileptic events are conformant to each other and therefore a suitable distance measure can cluster them apart from the non-epileptic false positive detections. Fundamental models of the epileptic neurological condition suggesting that seizures may be states representing limit cycle dynamics [11] back this working hypothesis.

First, we define the channel-average (we remind that channels are the group parameter rates) deviation or energy of the optical flow for each *q*-sequence as:(9)$$P\left(q\right)=\langle std_{t\in q}\left(V_{g}\left(t\right)\right)\rangle_{g}$$

Next, we take for each detected event i=1…M those frame sequences that are 20 windows (frame sequences) before and after the event. For the so obtained set of signals $P_{i}\left(q\right),i=1\ldots M,$ collected around each detected event, we define the distance between each pair as: (10)$$D_{ij}=min_{\tau }(d_{ij}\left(\tau \right));d_{ij}\left(\tau \right)=\frac{\sum_{q}\left|P_{i}\left(q+\tau \right)-P_{j}\left(q\right)\right|}{\sum_{q}\left|P_{i}\left(q+\tau \right)+P_{j}\left(q\right)\right|}$$

According to (10) all the distances are bounded between 0 and 1. The metric is therefore non-quadratic and allows for an easier parametrization of the clustering criterion. 

Foe each event the mean distance to all the other recorded events is: (11)$$C_{i}=\langle D_{ij}\rangle_{j}$$

The clustering of events defines seizure event category: (12)$$Q_{i}=(C_{i}\leq \varepsilon )$$

Here ε is a threshold parameter. We use here the convenient value for the threshold ε = 0.5. This value corresponds to the average distance between each event and the rest in case of randomly selected sequences. Another, more conservative choice is ε = 0.45 prevents classifying some false-positive detections as true ones. 

## 3. Results

### 3.1. Performance of the Default Algorythm

Under standard (default) parameter settings: [f1,f2] = [2,7] Hz; [Tr,N,n] = [0.4,7,6] the detection results from a long-term observational facility are shown on Figure 3. After manual validation of the system continuous operation for 230 days, the total number of detections was 228. Because the video recording was not always active, especially at the beginning of the trial due to delayed informed consent from the patient, also later due to corrupted or missing video files, 30 events were nor validated. The system records continuously video fragments of 20 min duration. We kept those video fragments containing detected events and deleted the rest to save storage capacity. 

From the validated 198 events, nine were deliberate tests and are not used to assess the algorithm. The remaining 189 events represent 132 TC seizures and 57 false positive detections of various origin. Therefore, we can conclude that the estimated specificity of the system is 100 × 132/189~70%. Considering the 30 non-validated detections, the range of the specificity can be extrapolated as 100 × [162, 132]/219~[60, 74]%. The false positive rate is 230/57~0.25 FP/day or inversely one false alarm per four days. In the worst-case scenario, when all non-validated events are false positives, those numbers are correspondingly 0.38 FP/day, or correspondingly, one FP per 2.6 days. 

### 3.2. Parameter Optimization after Supervised Validation

We applied parameter optimization using some of the detected events. From a sequence of 20 validated TC events, we extracted as seen on Figure 4, left frame an optimized frequency range—from 1.5 Hz to 5 Hz as candidate to replace the original 2Hz–7 Hz ad hoc detection interval. As indicated on the right panel the optimal detection parameters were suggested as [Tr,N,n] = [0.45,8,7] corresponding to the new frequency range. 

We changed the parameters and let the system continue its detection cycle uninterruptedly. Figure 5 shows the statistics from the following 36 days. Video recordings and visual validation applied to all detections. From 32 detections, we excluded the two tests and estimated the performance on the 30 true TC detections and two false positives. This left us with a specificity of 100 × 28/30 ~ 93% and false positives rate of 2/36~0.06 FP/day or one FP per 18 days. 

Finally, we investigated the stability of the adaptive process. In the period after the parameter adjustments, we applied again the parameter optimization procedure. The frequency range determined by the algorithm remained practically the same [1.5–5] Hz and the optimal detection parameters are shown on Figure 6. We see that the new optimal threshold is 0.42 (was 0.45) and the detection times are [9,8] instead of [8,7]. This shows a relatively robust adaptive process, and the minimal difference represents a tradeoff between the threshold value and the time interval of sustaining the feature above the threshold in order to generate an alert event. 

### 3.3. Unsupervised Event Validation

Our final set of results concerns the challenge to automate the validation procedure as proposed in Section 2.8. From a sequence of 66 detections, visual validation has concluded 52 TC events, 11 false positives and three deliberate seizure imitation tests by caregivers trying to trigger a bed-mounted seizure detection sensor (EpiCare^®^). We note here that the high number of false positives obtained within two successive days is due to frequent convulsive movements of the patient, which are also paroxysmal but are not TC seizures. For the Table 1 we see that excluding the test events, all but three out of 63 events are properly classified off-line resulting in 100 × 60/63~95% accuracy. The false positive detections were zero, and the false negative fraction, the percentage of TC events excluded by the clustering analysis is 100 ×3/52~6%, resulting in 94% sensitivity. 

The next result presents the outcome of an approximately two-month fully autonomous operation of the system in adaptive mode. Figure 7 summarizes both the detection quality and the assessment of automated retrospective classification used for the continuous parameter optimization. Note that the operator-based scores (blue bars) are only to validate the unsupervised procedure and have no consequences for the parameter adjustments. 

From the data of Figure 7, we conclude that the accuracy of the unsupervised classifier is 100%, excluding the test events. The overall specificity of the detection in this autonomous adaptive operational mode is 100 × 40/48 = 83%. The false positive rate is one FP/week. 

Results from Table 1 and Figure 7, show that our major hypothesis underlying the clustering technique is valid for these cases. The difference between the two sets is that the data in Table 1 are during a supervised parameter optimization procedure while the clustering method provides a proof of the concept. For the results presented on Figure 7, the clustering method was the running classification algorithm, and the visual score is for validation only. In both cases, true seizure events tend to cluster as opposed to the false positive events. This result indicates that, indeed, TC seizures follow specific patterns. 

## 4. Discussion

### 4.1. Features of the Proposed Method

Our concept is different from the “standard” machine-learning paradigm in that it accumulates its training set during the normal, ongoing operation of the system. There is no separation between the learning phase and the performance phase. When the system collects a sufficient number of events (we have selected a minimum of 10 in the current trial), the adaptation algorithm activates autonomously or by operator′s intervention. Our scheme thus avoids the need for large, validated data sets in order to train the detection algorithm in advance. Collecting such data, especially in cases of infrequent epileptic seizures, may require large time intervals before the system is ready to operate in real time. Another advantage is that in non-stationary situations, the adaptation can be perpetual and not restricted to an initial training. The autonomous classification can also restrict the number of previous events considered for the adaptation procedure. Events that served as a training set earlier may not be relevant later if the conditions (medication, environment) have changed. 

In our unsupervised validation approach, we use a multivariate description of the data as opposed to the feature-based detection algorithm employed for the real-time detection. In this way, the total amount of accumulated data drives the adaptive process. The more events in the training set; the better is the classification accuracy. Although the clustering technique is dependent of the events generated by the detection algorithm, the clustering quantifier described in Section 2.9, the mean distance to the rest of the events, is not related or dependent in any way on the processing parameters. Neither the frequency range nor the detection threshold can directly influence the result of the clustering. Therefore, it is possible to apply the automated labeling of the events on any subset of events regardless of the parameters used to detect them. 

### 4.2. Comments on the Results

The specificity of the detection algorithm, the ration between the true detections and the total number of detections, improves with the introduction of parameter adaptation. In the “rigid” mode, this feature was 70% while after introducing the on-going learning procedure it went above 83%. To compare with previous results in [3,5] where the detection was of a “rigid” type with only fixed parameters, we note that the reported false positive rates of 0.7–1.0 per night are higher than in our trial. In [5] the specificity is according to the sensitivity level, for 100% sensitivity the averaged specificity is at 80%. It is difficult however to compare the approaches as we have very limited amount of data. Another factor is the source of false alarms. In most other works they are reportedly due to behavioral events while in our case majority false positives are actually due to intermittent motor paroxysms during sleep that are not tonic-clonic seizures. 

Concerning the results from Table 1, the automated, unsupervised retrospective validation, we stress that the misclassifications of false negative type are not harmful for the adaptive concept proposed here. Unlike the missed detections during the online operation that may compromise patient′s safety, the missed true seizure events during the validation can only decrease the amount of events used for optimization of the detection parameters but in general will not alter essentially the outcome of the adaptive algorithm. On the contrary, FP type of misclassification may not be benign as it may introduce “alien” events into the training set and potentially steer the adaptive procedure away from optimal set of parameters. For the same reason we would favor test events being classified as false positives rather than as true TC seizures. We also decided to present that the outcomes of our second trial separately. Firstly, because it was the first trial that ran in fully automated adaptive mode, secondly and as stated in the previous sub-section, our scheme accounts for potential non-stationary situations. We limit the size of the accumulated training sets to a selectable number of previous events. 

### 4.3. Limitations 

Using “on the move” data to adjust detection parameters provides a dynamic reinforced learning scheme. However, it also contains a dependence on the initial state of the parameters. This creates the questions of convergence and stability, on the one hand, and missing the optimal settings on the other. So far, we have observed convergence and performance improvement, but in the future we will pursue more general evidence. 

Another limitation of the proposed scheme is the issue of false negatives, or missed true seizure events. Our current methodology relies only on detected events, thereby leaving possible undetected events out of the validation process. This is a generic challenge, as its solution may require scanning continuous video data by qualified observers. One of the obstacles in achieving a complete scan of the events is the necessity to continuously record large amounts of data. It is, however, clear that no method will ever give a 100% sensitivity. Even thorough visual inspection of long-time continuous video records by qualified observers cannot guarantee the absence of missed seizure events. 

For the automated classification, particularly dubious are those events where care personnel deliberately imitate convulsive seizures in order to test one or another alerting device. The automated classification algorithm will classify these “faked” seizures as either real ones or false positives depending on how realistic the imitation has been. In this respect, the visual operator-based validation will always be more accurate and therefore unequivocally serves as “ground truth”. 

Our approach essentially relies on the assumption that only the seizure events form a cluster. If other habitual movements are frequently present in the optical flow data, this may alter the clustering outcome. A more advanced multi-cluster analysis can address those cases. 

We have addressed here the major motor events that can have adverse effects for the patient. More subtle clinical events may go beyond the sensitivity range of pure video observation and require other sensors. This issue may certainly be relevant for diagnostic purposes. It remains, however, an open question whether all sorts of epileptic seizures can and should be detected and alerted for care purposes. 

### 4.4. Future Research 

One possibility for future research is to study the dependence of the adaptive algorithm on the initial state using data from multiple patients to derive an optimal set of parameters for the group. In such a scenario, group optimization will provide the initial setting of the personalization process and will improve with the inclusion of new patients. A secondary adaptive process will then take place on a personal level but starting from an already optimized initial state. We are now expanding the testing of the detection concept and we will be able to apply these ideas. To acquire more data from multiple subjects, we will use our pre-surgical observation facility where patients who are possible candidates for surgical treatment undergo video-EEG diagnostics while on controlled decrease of medication levels. The typical admission duration is between seven and 14 days, which is short for applying any adaptive algorithms. It is, however, possible to derive a group-optimal parameter set for later use in the residential facility as an initial state of the alerting system. 

We note also that the concept of remote detection of adverse events can go beyond the application of convulsive epileptic seizures. We have shown earlier that certain dynamic features of the seizure process can be predictive for post-ictal suppression of brain activity [12]. Falls and non-obstructive respiratory apnea can also be detected from video sequence analysis [13,14]. Upon proper testing and validation, the extended detection modules can be attached to the existing software and provide an enhanced automated “situation awareness” tool. 

Addressing the challenge of missed epileptic events, the automated off-line clustering approach is applicable to larger sets of retrospective data. This will produce a list of suspected events that the on-line detection cycle may have missed. We are currently investigating this technique and designing a recovery module that, if proven reliable, will be integrated into the existing system. 

Our approach is a “hybrid” one that combines model-based feature extraction from optical flow data with machine learning adaptive techniques. This gives several advantages when compared to pure machine learning concepts where large data sets must be prepared and scored in advance [15]. Extracting interpretable features does not only reduce the computational complexity of the classification task but also may provide valuable data for the diagnostics of the disease. 

## 5. Conclusions

In this short communication, we presented a novel methodology for automated, remote sensor detection of convulsive epileptic seizures. The base algorithm uses optical flow reconstruction followed by wavelet frequency filtering and detection criterion. The system works in real time, a USB day/night camera feeds the data, and after processing the alerts are sent to a nursing alerting system vie USB-controlled dry contacts. 

In addition to the rigid video sequence processing, an adaptive parameter optimization algorithm improves the performance of the detector. The procedure collects data during operation and does not require pre-recorded training set. This part of the system can use either event classification by visual inspection of the video records, or an automated classifier based on retrospective clustering analysis. In either case, a list with true alerts provides data for the machine learning modules. 

Besides real motor seizures and false positive detections, test events (typically deliberate seizure imitations by the personnel aimed at testing one or another detection device) were present in the trial as well. Operator-based validation excluded them from the training set. In the automated off-line detection, they can fall into either false or true epileptic events, depending on how close the testing procedure can imitate a real motor seizure.

Our final conclusion concerns the significance of the fully automated adaptive learning functionality proposed here. On one hand, it relieves the personnel of time-consuming validation procedures. In extra-mural applications where a qualified observer is not available, this may even be the only possible scenario for personalizing the system performance. Perhaps an even more important argument in favor of fully automated operation is that human-based visual inspection of video records may compromise a patient’s privacy. The automated classification introduced in this communication avoids this problem, as it does not require a recording of the raw video. It uses only the traces of the reconstructed group-velocities optical flow, which cannot reveal a patient’s identity. 

## Figures and Tables

**Figure 1 sensors-23-00968-f001:**
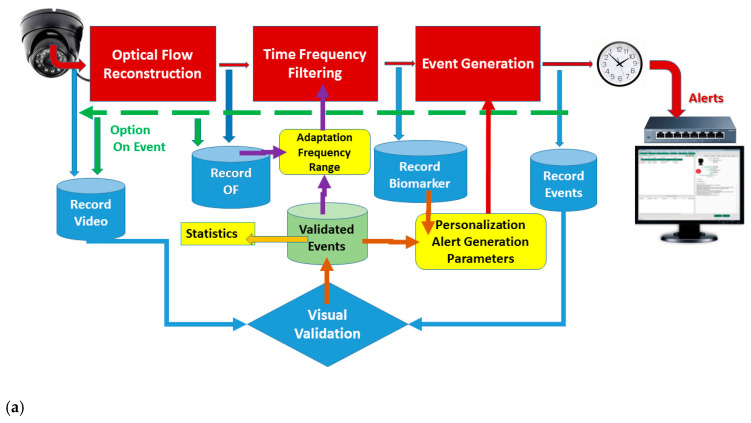
Overall processing flow of the proposed seizure detection and alerting system. USB day/night camera delivers fixed frame rate video stream to a PC platform. Optical flow algorithm for reconstruction of group parameter velocities and a set of Gabor wavelet decomposition and filtering computes the relative spectral content of the movements that corresponds to motor seizure. The so obtained biomarker generates an alert if the time above certain threshold exceeds pre-defined limits. The system sends the alerts, within selectable time schedule, to the nursing alert system through USB controlled latch relay. Additional “on demand” functionalities include recording intermediate data, including the input video sequence, optical flow, and the biomarker. The list of generated events always enters a log file. Validated events provide input for tuning the alert generation parameters and/or for adjusting the range of spectral components that identify epileptic seizures. On panel (**a**), the qualified observer does the validation off-line by inspecting the video sequences corresponding to the generated events. On panel (**b**), the scenario shown utilizes an additional algorithm for retrospective, cluster density-based classification. It uses data from the optical flow and does not require the raw video sequences. Recorded video fragments can provide if required human assessment and validation of the overall system performance but do not influence the parameter optimization process.

**Figure 2 sensors-23-00968-f002:**
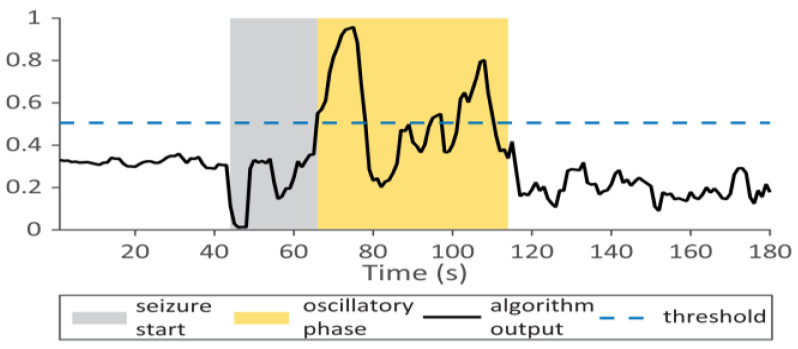
Illustration of the detection algorithm for seizure event. The horizontal axis shows the time or the sequence number. The vertical axis represents the epileptic marker (4). If the marker is occupying more than n points from the tested N, the algorithm generates a seizure event.

**Figure 3 sensors-23-00968-f003:**
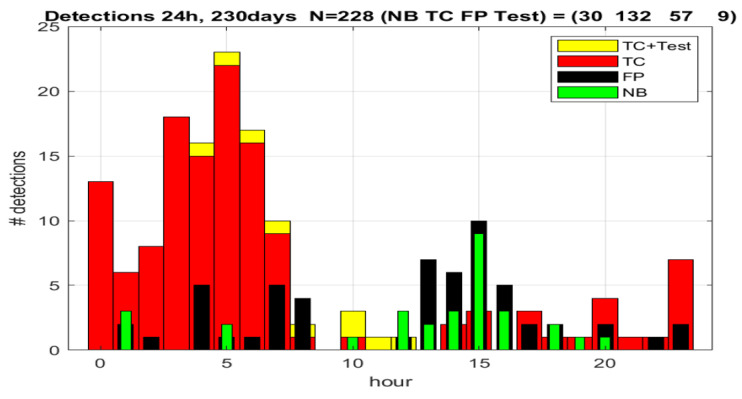
Detection statistics with default and ad hoc parameters. TC—tonic–clonic seizures; FP—false positive; Tests—deliberate seizure imitation; NB—no validation possible. Horizontal axis indicates the hour during 24-h detection summary. Vertical axis gives the numbers of the detected events.

**Figure 4 sensors-23-00968-f004:**
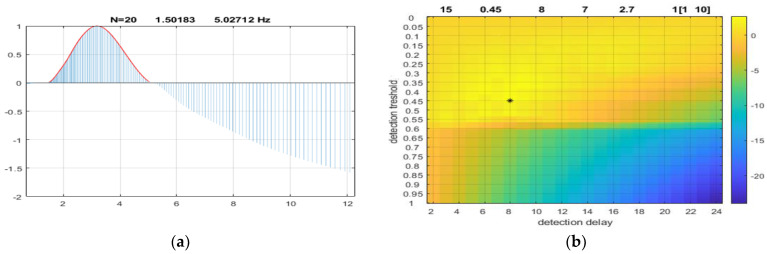
Results from parameter optimization technique. Panel (**a**): the proposed frequency range. The horizontal axis is the Gabor central frequency in Hz, the vertical is the relative corrected spectrum as defined by Equation (5); Panel (**b**): the optimal detection parameters. Vertical axis is the threshold measured as dimensionless quantity between zero and one. The horizontal axis represents the N-parameter, the detection delay measured in number of windows. The color code is the value of the cost-function given by Equation (7).

**Figure 5 sensors-23-00968-f005:**
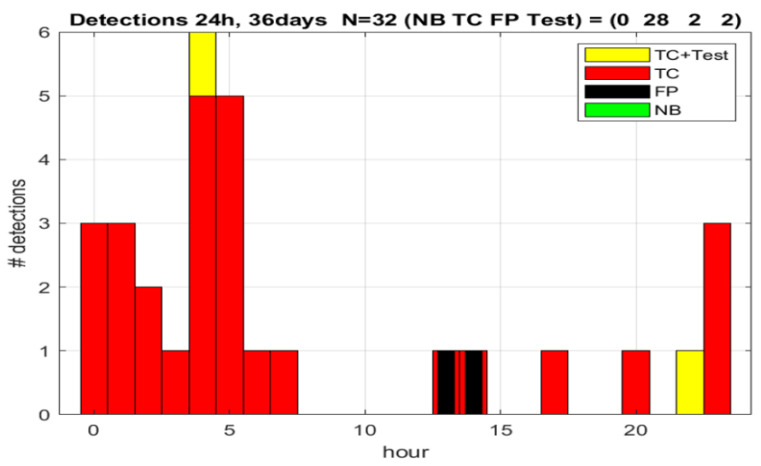
Detection statistics from optimized personal parameters (the same notations as on Figure 3).

**Figure 6 sensors-23-00968-f006:**
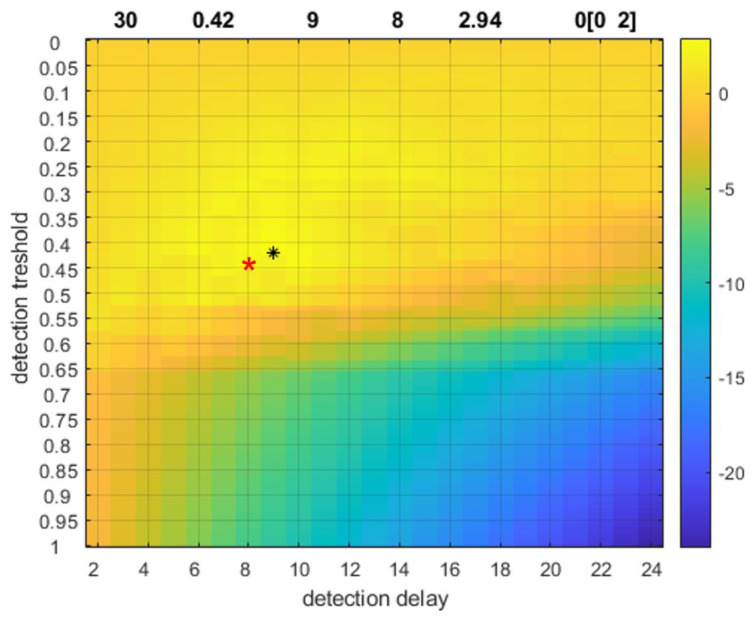
Stability of the adaptive approach. The notations are the same as on Figure 4, panel (b). Horizontal axis is the number of windows (each of approximately 1.5 s) and the vertical axis is the threshold. Both numbers are dimensionless units. The red star represents the initial optimal parameter combination, the black star and arrow the new proposed parameters by the algorithm.

**Figure 7 sensors-23-00968-f007:**
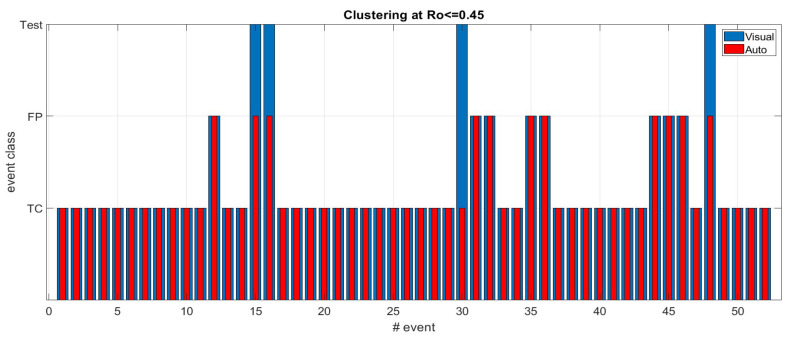
Outcome of fully autonomous system operation including unsupervised adaptive parameter optimization. Vertical axis denotes the event type; horizontal is the number of the event. The blue bars are the operator-validated event types, the red ones are those obtained by the clustering classifier. From 52 events, we have eight false positive detections, four test alerts and 40 tonic-clonic seizures. The algorithm classified correctly all 40 seizures as well as all eight false positives. Three out of the four test events obtained false positive classification, one was classified as a true seizure.

**Table 1 sensors-23-00968-t001:** Comparison between visual and automated event validation. We selected a session with 66 detected events. Visual review of the recorded video fragments revealed 52 TC seizures of the patient, 11 false alarms related to the condition of the patient but not TC seizures and three test events where the caregivers were testing another detection device working in parallel with the camera detection. The numbers are in the second column. The third column gives the classification according to the automated clustering algorithm. The numbers present the events classified as epileptic seizures. From the 52 validated seizures, 49 are correctly classified. The rest of the three are therefore false negative classifications. All the false alarms were outside the cluster criterion and therefore labeled correctly. From the three test events, one was interpreted by the system as a TC seizure, and the other two were outside the cluster and labeled as false positives.

Type Event	Numbers as Validated by Visual Inspection of The Video	Classified as Epileptic Events by the Clustering Criterion
TC seizure	52	49
False Alarm	11	0
Test	3	1

## Data Availability

Not applicable.

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
