# Peer review of "Adaptive Remote Sensing Paradigm for Real-Time Alerting of Convulsive Epileptic Seizures"

_sensors, 2023, doi:10.3390/s23020968_

Round 1
Reviewer 1 Report
This paper researches an adaptive remote sensing paradigm for real-time alerting of convulsive epileptic seizures, analyzes the research method and gives some validation. But, there are still some problems with the paper.
1.The author mentions that he has done research on a non-contact method using automated video camera observation and optical flow analysis undergoes field trials in clinical settings. May I ask what is the key technology of the major breakthrough in this paper? What are the innovations? Is the current research addressing the same questions as the published research?
2. From the perspective of the whole paper, the logic of the paper is not very novel. For example, in the part 2, every node is relatively simple and there is nothing outstanding. In addition, the part 2.4- 2.7, the theory is a very mature function. The theoretical mathematical method of this paper is relatively weak, and there is a lack of support for specific methods. It is suggested that the author combine his own research objects and give specific scientific theoretical methods.
3. The overall organization of the paper is missing in the introduction
4. The author should give the summary and conclusion of the literature review, and the overall organization of the paper is missing in the introduction.
5. A comparison table of your solution with existing solutions from the literature must be added.
6. There are two formula (7) serial numbers in the paper, the whole format is not standard.
7. The conclusions of Table 1 are hard to convince because of the small number of data sets.
8. In part 4, the author makes some discussion, but it is not organized clearly, and does not draw conclusions about specific research points from the methodology in Part II.
9. There are no obvious new methods related to sensor concept and design in this paper, so I think this paper fails to meet the standards for publication.
Author Response
This paper researches an adaptive remote sensing paradigm for real-time alerting of convulsive epileptic seizures, analyzes the research method and gives some validation. But, there are still some problems with the paper.
1.The author mentions that he has done research on a non-contact method using automated video camera observation and optical flow analysis undergoes field trials in clinical settings. May I ask what is the key technology of the major breakthrough in this paper? What are the innovations? Is the current research addressing the same questions as the published research?
We appreciate this request of the reviewer. Although we have stated briefly in the introduction the claimed novelties in this work, it has been perhaps too briefly and not clear enough. The main novelties proposed here are (1) Algorithm for continuous adaptation of the seizure detection parameters and (2) Clustering-based algorithm for retrospective event classification. Using these two techniques, the system is currently operating completely autonomous and is successfully adapting to the particular patient and environment. In addition, the work uses for first time the group parameter rates reconstruction algorithm (published earlier as analytical method) in real time application. In the revised manuscript, we have added the above information in the Introduction and in the Conclusions.
- From the perspective of the whole paper, the logic of the paper is not very novel. For example, in the part 2, every node is relatively simple and there is nothing outstanding. In addition, the part 2.4- 2.7, the theory is a very mature function. The theoretical mathematical method of this paper is relatively weak, and there is a lack of support for specific methods. It is suggested that the author combine his own research objects and give specific scientific theoretical methods.
The mathematical foundations of group-parameter optical flow reconstruction have been publisher earlier. To our knowledge, they contain solid mathematical methodology, group theory and inverse problems in particular. In essence, this approach surmounts the computationally expensive optical flow reconstruction in each image pixel and instead directly estimates the group parameter velocities. The mathematical apparatus used in the current work is indeed not the primary claim. Still the definition of bounded (non-quadratic thus) distances between events (eq. (7) in the original submission) is not a straightforward approach. We have added a comment on this in the revised version.
- The overall organization of the paper is missing in the introduction
Added as required
- The author should give the summary and conclusion of the literature review, and the overall organization of the paper is missing in the introduction.
As above, added
- A comparison table of your solution with existing solutions from the literature must be added.
We have added some comparison of the sensitivity and false positive rates published earlier. We mention also that rigorous comparison is difficult at this stage due to the restricted data set.
- There are two formula (7) serial numbers in the paper, the whole format is not standard.
Thank you for this remark. We have reworked the numbering of the formulae and tried to improve the layout.
- The conclusions of Table 1 are hard to convince because of the small number of data sets.
In the meantime, the data set has grown. It is still a single case study but now includes more events from a separate trial of fully autonomous adaptive operation. It is however worth noting that if the data set is too large, non-stationarity of the conditions can influence the outcome. For this reason, we divide the validation set into cells of certain selectable number of events. We have added a comment on this important issue in the Discussion.
- In part 4, the author makes some discussion, but it is not organized clearly, and does not draw conclusions about specific research points from the methodology in Part II.
We have completely reworked this part. Dividing the Discussion into four parts: features, results, limitations and future research, we believe the structure has been improved.
- There are no obvious new methods related to sensor concept and design in this paper, so I think this paper fails to meet the standards for publication.
Up to our knowledge, using optical video camera for non-contact detection of convulsive seizures introduces new methodology, even if the sensor as such is a standard, off-the-shell one. The major contribution here is the processing of the sensor data. It is of course up to the reviewers and Editorial Board to judge the suitability for the scope of the Journal.
Reviewer 2 Report
The authors present a method for remotely detection of convulsive epileptic seizures. The possible event of epileptic seizures is detected by monitoring the patient using RGB camera and implementing optical flow algorithm. The result of such event detection is a series of images which are saved. Using the event sequence, authors use wavelet filtering to classify the events and recognize an epileptic seizures event. One of major task of the paper is mentioned as to reduce the false positive alarm during epileptic seizures detection.
These are my comments:
· The performance of chosen optical flow in conjunction of event detection of epileptic seizures in form is totally ignored. The authors should show a typical performance measurement about it. The lack of understanding of such performance may have major effect on saving the necessary sequence of frames (as it happened during their long 230 days test).
· The authors used many fixed rigid values such as: saving about 10 second of monitoring each time, 0.4 threshold value, N as 7, n as 6, o.1 in equation 5, using 10 windows before and 15 windows after. I can understand that these values may be used as initial values but using the one level fix rigid thresholding is highly result to rigidity of the system. The results in Figure 3 strongly indicate my point. The authors should explain why they could not obtain more flexibility in the algorithm by using the statistical values in their dynamic data series.
· The authors due to rigidity of the algorithm (see above) needed the expertise opinion (called as supervised learning) to adjust the algorithm (called as adaptive learning). This results to obtain the automatic algorithm (as authors call it). In my opinion this is highly doubtful to call the algorithm as automatic. I have two reasons for it: a) even for the same patient in the paper it may new seizure signals can appear which are not seen in previous supervised classifications. b) other patients certainly have different signals and classification conditions which are adjustable with a certain expertise categorization (they can even have different parameterization space due to kind of seizures’ signals). Thus, I believe the authors may should call their automatic algorithm as semi-automatic and may explain the possibility to use the dynamic of their data in future to fully obtain an automatic system.
· The authors mention: “There is no sharp transition between a learning phase and a performance phase”. They mention this as the reason why like “standard” ML, they are not keen to accumulate their training. I believe it is not true. According to their rigid method it is almost impossible to accumulate the training. If they use statistical obtained parameters there would be plenty of opportunities to accumulate the statistical data and obtain more interesting models related to it.
Author Response
The authors present a method for remotely detection of convulsive epileptic seizures. The possible event of epileptic seizures is detected by monitoring the patient using RGB camera and implementing optical flow algorithm. The result of such event detection is a series of images which are saved. Using the event sequence, authors use wavelet filtering to classify the events and recognize an epileptic seizures event. One of major task of the paper is mentioned as to reduce the false positive alarm during epileptic seizures detection.
These are my comments:
- The performance of chosen optical flow in conjunction of event detection of epileptic seizures in form is totally ignored. The authors should show a typical performance measurement about it. The lack of understanding of such performance may have major effect on saving the necessary sequence of frames (as it happened during their long 230 days test).
The performance of the non-adaptive or “rigid” algorithm applied on pre-recorded video sequences has been reported in earlier publications. In this work, we present also the current performance data in figure 3. What we are not able to show in this contribution is the statistics of false negatives, or missed seizures. Although we have mentioned this limitation in the original version, we have made it more pronounced in the revised version.
- The authors used many fixed rigid values such as: saving about 10 second of monitoring each time, 0.4 threshold value, N as 7, n as 6, o.1 in equation 5, using 10 windows before and 15 windows after. I can understand that these values may be used as initial values but using the one level fix rigid thresholding is highly result to rigidity of the system. The results in Figure 3 strongly indicate my point. The authors should explain why they could not obtain more flexibility in the algorithm by using the statistical values in their dynamic data series.
This is an important question; we thank the reviewer for bringing it up. The system was first used in rigid setting or the parameters (thresholds and time windows) were adapted “ad hoc” by operator. This arbitrariness has motivated the present study where we use adaptive paradigm that aims at parameter tuning according to supervised or autonomous validation of the detections. While there are still some, we believe non-critical, choices of mainly time intervals, the overall detection flow now has gained flexibility in (1) frequency range for the paroxysmal activity and (2) decision parameters for raising alerts. The fixed values are then only used as initial state that further adapts to the conditions. We added the above summary to the Introduction and the Discussion.
- The authors due to rigidity of the algorithm (see above) needed the expertise opinion (called as supervised learning) to adjust the algorithm (called as adaptive learning). This results to obtain the automatic algorithm (as authors call it). In my opinion this is highly doubtful to call the algorithm as automatic. I have two reasons for it: a) even for the same patient in the paper it may new seizure signals can appear which are not seen in previous supervised classifications. b) other patients certainly have different signals and classification conditions which are adjustable with a certain expertise categorization (they can even have different parameterization space due to kind of seizures’ signals). Thus, I believe the authors may should call their automatic algorithm as semi-automatic and may explain the possibility to use the dynamic of their data in future to fully obtain an automatic system.
Please see also the previous answer. The algorithm we propose is in two versions. The semi-autonomous (or semi-interactive) uses operator’s validation to adjust the parameters. This adjustment is in fact of statistical nature (equation (4) for example) and it uses the accumulated data from the previous detections. The second version uses however fully automated cluster-based classification of the previously detected events and accordingly adapts the detection parameters. The operation of the system in this mode runs completely autonomously and adapts the parameters during its operation. This said, we agree with the reviewer’s remark that in some cases seizures may be undetected from the beginning and therefore will fall outside the adaptation procedures, either supervised or automated. This remains an open challenge that is being addressed and will be reported shortly. We have commented on this in the original manuscript but in the revised version have added a more extensive discussion. We also added a second set of results (figure 7 in the revised manuscript) that represents the fully automated performance.
- The authors mention: “There is no sharp transition between a learning phase and a performance phase”. They mention this as the reason why like “standard” ML, they are not keen to accumulate their training. I believe it is not true. According to their rigid method it is almost impossible to accumulate the training. If they use statistical obtained parameters there would be plenty of opportunities to accumulate the statistical data and obtain more interesting models related to it.
Here there is possibly a misunderstanding, we apologies for the confusion. What we meant is that our approach does not require preliminary collection of labeled data. Instead, it collects the data while performing the detection task in real time. In the autonomous operational mode, the algorithm re-evaluates with each new event the optimal parameters according to the criteria presented in the text. We have clarified this in the Methods stating that there is no preliminary learning phase to be followed operational one. We also modified and added accordingly comments in the Discussion and the Conclusions.
Reviewer 3 Report
The author notes that besides real motor seizures and false positives, test events (typically deliberate seizure imitations by the personnel aimed at testing one or another detection device) were present. Operator-based validation excludes them from the training set. In automated off-line detection, they can fall into either false or true epileptic events depending on how close the testing procedure can imitate real motor seizure.
As a specialist doctor treating patients suffering from various types of epilepsy, I would stress that detection of motor seizures is a relatively simple job. However, temporal lobe epilepsy or other focal epileptic fits are extremely difficult to discover. For management practice, these latter forms of epileptic disease are much more critical.
Author Response
Comments and Suggestions for Authors
The author notes that besides real motor seizures and false positives, test events (typically deliberate seizure imitations by the personnel aimed at testing one or another detection device) were present. Operator-based validation excludes them from the training set. In automated off-line detection, they can fall into either false or true epileptic events depending on how close the testing procedure can imitate real motor seizure.
This is correct. The automated method cantor classify the test events as such so they fall into either true or false positive detections. This will make the operator-based validation always the more precise one; it is also accepted throughout the work as “ground truth”. One of the goals of our endeavor is to achieve automated classification as close as possible to the visual one. We have added this remark to the discussion.
As a specialist doctor treating patients suffering from various types of epilepsy, I would stress that detection of motor seizures is a relatively simple job. However, temporal lobe epilepsy or other focal epileptic fits are extremely difficult to discover. For management practice, these latter forms of epileptic disease are much more critical.
Indeed, we have concentrated in this development on convulsive motor seizures. In our understanding, those are the seizures with high risk of ictal and post-ictal complications. Although they are relatively easy to recognize, an automated 24/7 alerting can be still a challenge. Most of the available products are based on contact sensors that are not suitable in all cases. Moreover, as the reviewer properly mentions, other types of epileptic fits, even generalized ones but without motor paroxysms (absences for example) are hard to detect, especially without the aid of EEG monitoring. We admit this limitation of our approach in the Discussion. We also note there that not all types of epileptic seizures require assistance or alerting although their detection may still be important for diagnostic purposes.
Round 2
Reviewer 1 Report
This paper has been modified by the author, the expression and statement of the paper are clear, and the corresponding explanations are added and supplemented. The current paper is reasonable, But it needs to be revised before it can be published
(1) there are also some small problems. The text in Figure 1,Figure 4 and 7 are not clear. The text in the picture is hardly legible. It is suggested to give a clearer picture.
(2) There is no corresponding unit name in Figure 6, and the figure is not described clearly, How to illustrate the advantages of the new method? It is suggested that the author give a supplementary note that the comparative results cannot be given by just black star and arrow, which is obviously insufficient. Figure 6 is very unscientific.
(3) In the verification of the part 3, the relationship between the calculation of this part and the previous theory is still not seen.The author is advised to give an appropriate explanation for this part.
